# Dietary Advanced Glycation Endproducts Induce an Inflammatory Response in Human Macrophages in Vitro

**DOI:** 10.3390/nu10121868

**Published:** 2018-12-02

**Authors:** Timme van der Lugt, Antje R. Weseler, Wouter A. Gebbink, Misha F. Vrolijk, Antoon Opperhuizen, Aalt Bast

**Affiliations:** 1Department of Pharmacology and Toxicology, Faculty of Health, Medicine, and Life Sciences, Maastricht University, 6229 ER Maastricht, The Netherlands; a.weseler@maastrichtuniversity.nl (A.R.W.); antoon.opperhuizen@maastrichtuniversity.nl (A.O.); a.bast@maastrichtuniversity.nl (A.B.); 2RIKILT, Wageningen University and Research, 6708 WB Wageningen, The Netherlands; wouter.gebbink@wur.nl; 3Campus Venlo, Maastricht University, 5911 AA Venlo, The Netherlands; m.vrolijk@maastrichtuniversity.nl; 4Office for Risk Assessment and Research (BuRO), Dutch Food and Consumer Safety Authority, NVWA, 3511 GG Utrecht, The Netherlands

**Keywords:** advanced glycation endproducts, inflammation, food, macrophages, Maillard reaction

## Abstract

Advanced glycation endproducts (AGEs) can be found in protein- and sugar-rich food products processed at high temperatures, which make up a vast amount of the Western diet. The effect of AGE-rich food products on human health is not yet clear and controversy still exists due to possible contamination of samples with endotoxin and the use of endogenous formed AGEs. AGEs occur in food products, both as protein-bound and individual molecules. Which form exactly induces a pro-inflammatory effect is also unknown. In this study, we exposed human macrophage-like cells to dietary AGEs, both in a protein matrix and individual AGEs. It was ensured that all samples did not contain endotoxin concentrations > 0.06 EU/mL. The dietary AGEs induced TNF-alpha secretion of human macrophage-like cells. This effect was decreased by the addition of N(ε)-carboxymethyllysine (CML)-antibodies or a receptor for advanced glycation endproducts (RAGE) antagonist. None of the individual AGEs induce any TNF-alpha, indicating that AGEs should be bound to proteins to exert an inflammatory reaction. These findings show that dietary AGEs directly stimulate the inflammatory response of human innate immune cells and help us define the risk of regular consumption of AGE-rich food products on human health.

## 1. Introduction

During heating of food products containing protein and carbohydrates, dietary advanced glycation endproducts (AGEs) are formed via the Maillard reaction [1]. AGEs are greatly responsible for the flavor and taste of food, and increase palatability. In the Maillard reaction, the carbonyl group of a reducing sugar (sugars with a free anomeric carbon [2]) reacts with an amino moiety of an amino acid leading to reactive intermediates, the so-called Amadori products. These Amadori products then form different AGEs [3]. AGEs comprise a large body of different molecules [3]. Examples of well-studied AGEs are: N(ε)-carboxymethyllysine (CML), carboxyethyl-lysine (CEL), methylglyoxal-derived hydroimidazolone (MG-H1), pentosidine, and acrylamide. Although the quantification of AGEs in food products is frequently done by antibody-based techniques [4,5], liquid chromatography-tandem mass spectrometry (LC-MS/MS) is the preferred technique to accurately identify and quantify individual AGEs [1,6,7,8]. Physiological consequences of exposure to dietary AGEs are not yet fully understood. Endogenously formed AGEs in diabetic patients, however, have been shown to induce an inflammatory reaction and contribute to the onset of cardiovascular diseases, such as atherosclerosis and diabetic cardiomyopathy [9]. The inflammatory effect seen in these situations is proposed to be caused by the binding of AGEs to the specific receptor for advanced glycation endproducts (RAGE). RAGE activates many enzymes and protein complexes, one of which is nuclear factor kappa-light-chain-enhancer of activated B cells (NF-κB) [10]. Even though inflammation caused by endogenous AGEs has been thoroughly studied, information on the generation of inflammation by dietary AGEs is scarce. As said, many of the studies on the effects of AGEs are conducted with endogenously formed AGEs, by heating bovine serum albumin or human serum albumin with different sugars at 37 °C for several hours, mimicking the endogenous situation. This difference with a home cooking situation, in which dietary proteins and sugars are heated to higher temperatures, can lead to a different array of molecules, resulting in different immunological outcomes. In addition to endogenous AGEs, some individual AGEs have been researched. Acrylamide is a well-studied compound, but only with regards to its genotoxic and carcinogenic effects [11,12]. To our knowledge, no study has been published on the inflammatory effect of acrylamide. A clear cause-effect relationship between AGEs and inflammation has also not be established yet and the available results have been questioned as being caused by endotoxins rather than AGEs [13,14]. To our knowledge, we are the first to investigate the inflammatory effect of dietary AGEs in a home-cooking environment and the first to investigate the inflammatory effect of acrylamide.

The present study aimed at investigating the effects of dietary AGEs on the inflammatory response of human cells of the innate immune system. To mimic a home cooking situation, only dietary protein and sugars were used and heated to a temperature of 100 °C. The formation of dietary AGEs over time was quantified by LC-MS/MS and the presence of endotoxin was excluded. We also examined the effect of individual AGEs and whether the observed inflammatory effect was induced by activation of RAGE.

## 2. Materials and Methods 

### 2.1. Chemicals and Reagents

Casein from bovine milk, α-lactose monohydrate, NaOH, sodium-phospate, 2-mercaptoethanol, and thiazolyl blue tetrazolium bromide (MTT) were obtained from Sigma-Aldrich (Saint Louis, MO USA). D-glucose, glutamate, fetal bovine serum (FBS), Dulbeccco’s Phospate-Buffered Saline (DPBS) were obtained from Gibco (Thermo Scientific, Waltham, MA, USA). Analytical standards of CML (>99%), CEL (>95%), MG-H1 (>93%), and pentosidine (>99%), as well as the deuterium labelled internal standards CML-d2, CEL-d4, and MG-H1-d3, were obtained from Polypeptide (Strasbourg, France). Boric acid (99.5%), chloroform (99.5%), nonafluoropentanoic acid (NFPA; 99%), sodium hydroxide (98%), sodium borohydride (96%), trifluoroacetic acid (TFA; 99%), and phorbol 12-myristate 13-acetate (PMA) were obtained from Sigma (Zwijndrecht, Netherlands). HPLC-grade acetonitrile and methanol were obtained from Actu-all Chemicals (Oss, Netherlands). 

### 2.2. Preparation of Glycated Casein

Casein, glucose, and lactose, in the proportions of milk powder (11 mM glucose, 0.2 M lactose, 10 g/L casein from bovine milk), were diluted in 50 mM phosphate buffer, pH 7.4 in and heated in an Erlenmeyer on a heating plate at 100 °C up to 120 min. At time points 0, 15, 30, 60, 90 and 120 min. samples were taken and immediately cooled in ice water. Samples were aliquoted and stored at −80 °C.

### 2.3. Determination of Endotoxin in Glycated Casein

The presence of endotoxin in the glycated casein samples from all time points was assessed by the commercially available PYROGENT Gel Clot LAL Assay 0.06 EU/mL sensitivity (Lonza, Basel, Switzerland). The assay was performed in accordance with the manufacturer’s protocol using endotoxin-free dilutions and reagent tubes (Lonza, Basel, Switzerland).

### 2.4. Assessment of AGE-Formation by Fluorescence Spectroscopy

AGE-formation was monitored by fluorometry measuring AGE-specific fluorescence. 100 μL of each sample was added in duplicate in a clear bottom black polystyrene microplate (Corning, Corning, NY, USA) and read in a microplate reader (Spectramax M2 Multi-Mode, Molecular Devices, San Jose, CA, USA) at an excitation wavelength of λ_ex_ = 370 nm and an emission wavelength of λ_em_ = 440 nm [14].

### 2.5. Quantification of AGEs in Glycated Casein by UPLC-Tandem MS/MS

Sample preparation is based on published methods [6,15]. Briefly, sodium borate (0.33 M; pH 9) and sodium borohydride (2 M) were added to an aliquot of the samples (equivalent to ~1 mg protein) to obtain a sodium borohydride concentration of 0.2 M in the samples. These sample solutions were incubated for 4 h at room temperature. Subsequently, 1 mL of a chloroform/methanol (2/1 (*v/v*)) mix was added, and the samples were centrifuged for 10 min at 12,000 rpm. The chloroform phase was discarded and the remaining solution containing proteins was hydrolyzed in 6 M HCl at 110 °C for 18 h. The solutions were evaporated until dryness at 50 °C under a stream of nitrogen and dissolved in 1 *v/v*% aqueous TFA (1 mL). Prior to UPLC-MS/MS analysis, internal standards and MilliQ were added to 50 µL of these analyte solutions to achieve a final volume of 200 µL.

The samples were analyzed using an Acquity UPLC system (Waters, Milford, MA, USA) equipped with a BEH C18 analytical column (50 × 2.1 mm, 1.7 μm particle size, Waters, Milford, MA, USA)). The mobile phases were 5 mM NFPA in water (A) and acetonitrile (B), and the flow was set at 0.3 mL/min. The mobile phase gradient was as follows: a linear increase from 90% A (*t* = 0 min) to 50% A (*t* = 4 min.) followed by a 1 min. equilibration at the initial conditions. The column was kept at a constant temperature of 45 °C. A Quattro Premier triple quadrupole mass spectrometer (Waters) was connected to the UPLC was operated in positive electrospray ionization (ESI+). The capillary voltage was set to 3.0 kV, and the source and desolvation temperatures were 120 °C and 450 °C, respectively. The cone and desolvation gas flow were set at 100 L/h and 600 L/h, respectively. Compound-specific cone voltages and collision energies can be found in Appendix A. 

Quantification was performed using an internal standard approach and nine-point calibration curves. MG-H1-d3 was used as internal standard for the quantification of pentosidine. Quantification was performed using the precursor−product ion multiple reaction monitoring (MRM) transitions reported in Appendix A. Instrumental detection and quantitation limits (IDL and IQL, respectively) were determined as the concentration from a peak with a signal-to-noise ratio of 3 and 10, respectively, in a standard chromatogram. Method detection and quantitation limits (MDL and MQL, respectively) were determined for AGEs detected in sample extracts as the concentration from a peak with a signal-to-noise ratio of 3 and 10, respectively, in a sample chromatogram. For pentosidine (not detected in the samples) the MDL and MQL were determined using a value of 3 and 10 times the noise at the retention time of pentosidine as peak area. Compound-specific detection and quantification limits are listed in Appendix A. The arithmetic mean recovery (± SE) was 113 ± 2% for CML-d2, 104 ± 2% for CEL-d4, and 33 ± 1% for MG-H1-d3. The relative differences observed between duplicate analyses were 12%, 7%, and 15% for CML, CEL, and MG-H1, respectively.

### 2.6. Cell Culture and Exposure 

THP-1 monocytes (ATCC, TIB- 202), cultured in RPMI 1640 with L-glutamine, Hepes and phenol red (Gibco, Thermo Scientific, Waltham, MA, USA) supplemented with 10% (*v/v*) FBS, D-glucose (4.5 g/L), Na-pyruvate (1mM) and 2-mercaptoethanol (50μM), were seeded in a 96-wells plate at a cell density of 70,000 cells/well and differentiated into macrophages by adding 200 nM of PMA to the cell culture medium and culturing them for 72 h. After differentiation, the cells were exposed to 1, 2.5, 5 and 10% (*v/v*) of glycated casein in serum and phenol-red free medium and always in a 50% DPBS solution. Only samples that were proven to be free of endotoxins were used. Control conditions included phenol-red free medium and 50% DPBS. Incubations were done both with and without the anti-carboxymethyl lysine (CML) antibody (Abcam, Cambridge, MA, USA) in a 1/500 dilution. Samples with anti-CML antibody were incubated overnight at 4 °C before using. For the RAGE antagonist experiments, cells were incubated with 10% glycated casein and with and without 1 µM FPS-ZM1 (Merck, Darmstadt, Germany).

### 2.7. Assessment of Cell Viability

Cell viability was assessed using the MTT assay. Supernatant was removed and stored for ELISA analyses. 100 μL of MTT solution (0.5 mg/mL in DPBS) was added to each well, after which the plate was incubated in the dark at 37 °C for 1 h. After 1 hour, the MTT solution was removed from the wells and 100 μl of DMSO was added to each well. After 10 min incubation at room temperature, absorbance was measured at λ = 540 nm in a microplate reader (Bio-Rad, Hercules, California, USA).

### 2.8. Quantification of Tumor Necrosis Factor (TNF)—Release by ELISA

Tumor necrosis factor (TNF)-α release in cell culture supernatant was assessed by a commercially available ELISA kit (Sanquin, Amsterdam, the Netherlands). The analyses were carried out in accordance with the manufacturer’s protocol using the Pelikine Toolset for all reagents (Sanquin). For the individual AGEs and RAGE experiments, TNF-α release was assessed by a commercially available ELISA kit (R&D systems, Minneapolis, MN, USA). The analyses were carried out in accordance with the manufacturer’s protocol.

### 2.9. Statistics

The experiments were run in duplicate and triplicate on different days using the average of the duplicates as one value. The data were analyzed using the GraphPad Prism software (v 5.00, GraphPad Software, San Diego, CA, USA), tested for normality using the D’Agostino & Pearson omnibus normality test, and examined with the non-parametric one-way ANOVA Kruskal-Wallis test followed by Dunn’s Multiple Comparison Test or Mann-Whitney U test to compare the two sets of treatment. For comparing with controls set to 100%, the Wilcoxon signed-rank test was used. Significance level was set to *p* < 0.05. Significance is indicated as: * = *p* < 0.05; ** = *p* < 0.01; *** = *p* < 0.0013.

## 3. Results

### 3.1. Analytical Characterization of Glycated Casein

To assess the effects of AGEs on human macrophage-like cells, AGEs were made in the form of glycated casein, combining casein, lactose, and glucose. In order to monitor the formation of AGEs in the glycated casein, AGE-specific fluorescence was recorded during the heating process. The fluorescence signal increased until 90 min of heating. AGE-specific fluorescence did not further increase between 90 min to 120 min of heating (Figure 1).

CML, CEL, MG-H1 and pentosidine were quantified in glycated casein using UPLC-MS/MS. Pentosidine was undetectable in the samples. The formation of CML, CEL and MG-H1 in glycated casein during the 120-min heating period is shown in Figure 2. The concentrations of all three compounds increased with time of heating. At 15 min of heating, MG-H1 was present in concentrations of 1.6 ± 0.9 μg/mL (mean ± standard deviation (SD)), being slightly higher than 0.6 ± 0.5 μg/ml CML although not significantly. Between 60 and 120 minutes of heating, CML concentrations increased from 4.3 ± 0.8 μg/mL to 11.3 ± 1.7 μg/mL, making CML the most abundant AGE present in glycated casein after 2 h heating. CEL concentrations increased as well, but were an order of magnitude lower than CML and reached concentrations of 0.3 ± 0.2μg/mL after 120 min of cooking. 

### 3.2. Effects of Glycated Casein on TNF-α Release of Human Macrophage-Like Cells

In order to assess the inflammatory effects of glycated casein on human macrophage-like cells, THP-1 cells were exposed to glycated casein heated for 15 min. Six-hours exposure of THP-1 cells to various concentrations of glycated casein elevated the release of TNF-α in a concentration-dependent manner (Figure 3). Increasing the glycated casein concentration from 1 to 2.5% (*v/v*) led to an increase of 277 ± 170% TNF-α secretion (mean ± SD), whereas the increase from 5 to 10% (*v/v*) resulted in an increase of 141 ± 87%. Due to large standard deviations, only the increase of concentration from 1 to 10% glycated casein led to a significant increase of TNF-α secretion, 1584 ± 604% (*p* < 0.01).

In Figure 1 and Figure 2, an increase in AGE formation during prolonged heating at 100 °C is shown. To explore the effects of glycated casein on the viability of macrophage-like cells at different time points during the heating process, the cells were exposed to a 10% (*v/v*) solution for 6 h. Cell viability did not alter with increasing cooking time until 120 min of heating, at which the cell viability significantly decreased from 118 ± 11% to 86 ± 21% (Figure 4).

To investigate the influence of prolonged heating on the pro-inflammatory effects of glycated casein, we measured the TNF-α release of macrophages exposed for 6 h to glycated casein (1% (*v/v*)), which was heated for 15–120 min. As Figure 5 shows, TNF- α secretion did not change significantly between the different heating time intervals. There seems to be an increasing trend, but standard deviations are too large to draw that conclusion. Before adding the glycated casein to the cells, all samples were tested for endotoxin presence; the kit sensitivity was 0.06 EU/mL. No endotoxin was detected above these levels.

### 3.3. Effects of the AGEs in Glycated Casein on TNF-α Release of Human Macrophage-Like Cells

To confirm that the inflammatory reaction was caused by AGEs, an anti-CML antibody was added to mitigate the TNF-α secretion. Adding the anti-CML antibody to cells exposed to 2.5% (*v/v*), glycated casein reduced TNF-α secretion by 36 ± 24% (*p* = 0.1, Figure 6). Next to this, the cells were incubated with 10% (*v/v*) glycated casein with and without 1 µM of the RAGE antagonist FPS-ZM1. Addition of 1 µM of FPS-ZM1 led to a significant 30% decrease of TNF-α secretion. 

### 3.4. Effects of Individual AGEs on TNF-α Release of Human Macrophage-Like Cells

The cells were incubated with CML, CEL, MG-H1, and acrylamide in different conditions for 6 h to assess the effect of individual AGEs on TNF-α release. All compounds were tested in concentrations ranging from 0.1 mM to 3 mM. None of the individual compounds led to any TNF-α secretion by the macrophage-like cells (data not shown).

## 4. Discussion

In this study, we showed that dietary AGEs directly led to a concentration-dependent increase in TNF-α secretion from human macrophage-like cells, which is induced through the binding to RAGE.

Our AGEs were made in a home cooking environment at 100 °C and are therefore much more representative for dietary AGEs formed during food processing and cooking than AGEs formed at 37 °C. The presence of AGEs in the glycated casein was confirmed by UPLC-MS/MS, and testing the glycated casein for endotoxin presence eliminated the possibility of the TNF-α secretion to be induced by endotoxins. Exposing macrophage-like cells to the glycated casein induced a clear TNF-α secretion. This opposes the findings of Buetler et al., where eliminating endotoxins by TX-114 extraction diminished the capability of their solutions to induce TNF-α secretion in human cell lines [13]. Further differences between our study and that of Buetler et al., and others, is firstly their use of endogenous AGEs developed at 38 °C to our dietary AGEs, which are developed at 100 °C and secondly their use of human lung epithelial cells and human retinal pigment epithelial cell line opposed to our human immune cells. Immune cells are naturally much more prone to excrete cytokines. Interestingly, Cai et al. fractioned AGEs from food products and showed their involvement in intracellular oxidative stress and some inflammatory potential in human umbilical vein endothelial cells. This underlines our results that dietary AGEs induce cellular stress [16].

The glycated casein induced TNF- α secretion could be attenuated by anti-CML antibodies, showing that the AGEs directly induce TNF- α secretion. Addition of the anti-CML antibody did, however, not completely diminish the TNF- α secretion. Since CML is not the only AGE present in the glycated casein, this finding indicates that other AGEs can also exert an inflammatory effect. Our observation that only adding CML antibodies can mitigate the inflammatory effect induced by glycated casein is quite remarkable. Blocking RAGE with the antagonist FPS-ZM1 showed that the effect was caused by AGEs binding to RAGE. These combined results corroborate our conclusion that the TNF- α secretion is induced by AGEs specifically. The most well-known pathway for TNF-α production through RAGE stimulation is NF-κB activation via activation of mitogen-activated protein (MAP) kinases, leading to TNF-α secretion [17]. Indeed, this has been shown by Lander et al. in rat pulmonary smooth muscle cells. It would be of interest to investigate this in human macrophages in future experiments.

Opposed to our results from glycated casein, the individual AGEs CML, CEL, MG-H1 and acrylamide did not show any induction of TNF-α release in human macrophage-like cells. This result indicates that AGEs need to be bound to proteins to exert an inflammatory reaction. This hypothesis has been mentioned in multiple studies [18,19]. Remarkably, a lot of conflicting research exists on whether individual AGEs can bind to RAGE and thereby induce an inflammatory response and whether they need to be bound to proteins to have this effect. Our results underline the hypothesis that AGEs can exert an inflammatory reaction through binding to RAGE, but only when bound to proteins. The reason why individual AGEs do not bind to RAGE has to be investigated further.

What would the results of this study imply for the human situation? Endogenously generated AGEs are known to induce significant adverse health effects when formed in the human body at 37 °C. AGEs can bind to RAGE, thereby activating an intracellular signaling cascade ending in the excretion of pro-inflammatory cytokines, leading to metabolic diseases such as atherosclerosis [10,20,21]. Another important mechanism by which AGEs can cause damage to the human body is by crosslinking with endogenous proteins such as collagen and elastin and thereby increasing stiffness in, for instance, muscle tissue [22]. In this study, we showed that dietary AGEs can also induce pro-inflammatory cytokine secretion. If AGEs in food are able to enter the human body, they might be able to contribute to the body’s endogenous AGE pool and worsen specific metabolic diseases. Food products that contain AGEs are those high in proteins and sugars and are processed at high temperatures, such as baked meats, cereals, peanut butter, and chocolate [6,7]. It is currently discussed whether dietary AGEs are absorbed in the gastrointestinal tract and contribute to the endogenous AGE pool [23,24,25,26]. Upon absorption, these dietary AGE products may aggravate metabolic diseases by the production of pro-inflammatory cytokines, as our study shows. Independently from this question of absorption, it should be emphasized that the pro-inflammatory effects of AGE may also be of local relevance in the gastrointestinal tract and might contribute to the symptoms of diseases such as inflammatory bowel diseases, Crohn’s disease, and ulcerative colitis. 

## 5. Conclusions

In conclusion, our study revealed that dietary AGEs themselves stimulate TNF-α secretion in human macrophage-like cells. To what extent dietary AGEs pose a risk for developing inflammatory diseases in the human GI tract remains to be investigated.

## Figures and Tables

**Figure 1 nutrients-10-01868-f001:**
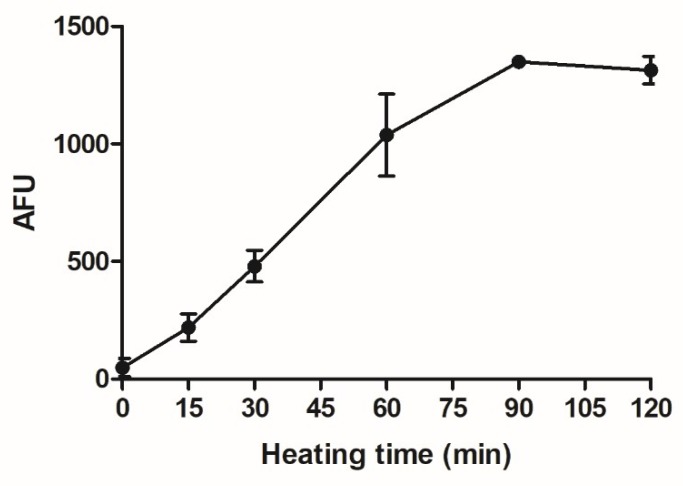
AGE-specific fluorescence of glycated casein measured at λ_ex_ = 370 nm/λ_em_ = 440 nm in relation to the heating time of the three components casein from bovine milk (10 g/L), lactose (0.2 M) and glucose (11 mM). Values were corrected for baseline fluorescence and data are shown as mean ± SD, *n* = 3.

**Figure 2 nutrients-10-01868-f002:**
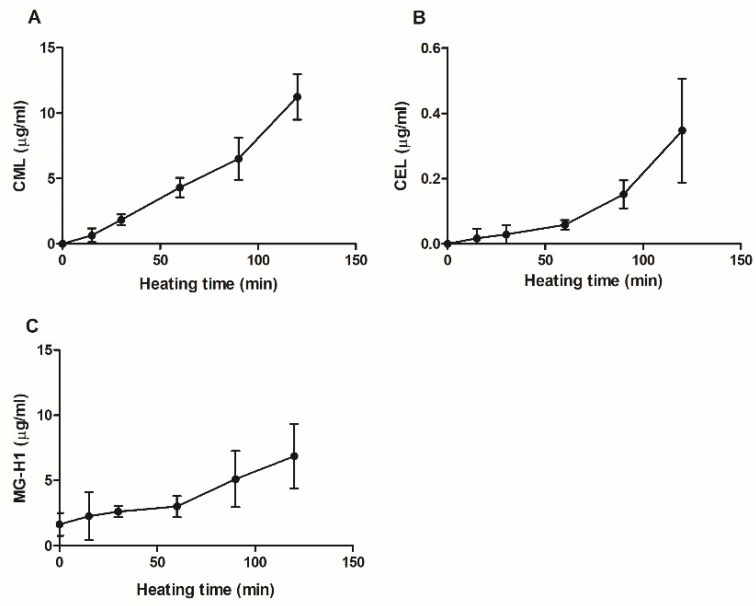
AGE-specific fluorescence of glycated casein measured at λ_ex_ = 370 nm/λ_em_ = 440 nm in relation to the heating time of the three components casein from bovine milk (10 g/L), lactose (0.2 M) and glucose (11 mM). Values were corrected for baseline fluorescence; the data are shown as mean ± SD, *n* = 3.

**Figure 3 nutrients-10-01868-f003:**
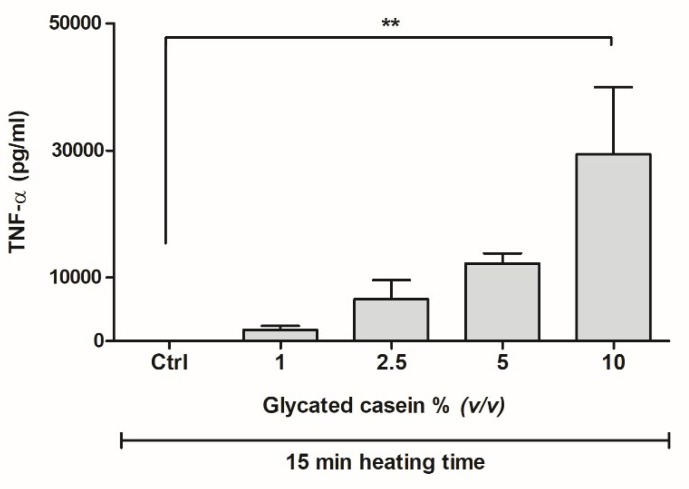
TNF-α secretion of human macrophage-like cells after 6 h exposure to different concentrations of glycated casein heated for 15 min measured by ELISA. Data are presented as mean ± SD, *n* = 3, ** = *p* < 0.01.

**Figure 4 nutrients-10-01868-f004:**
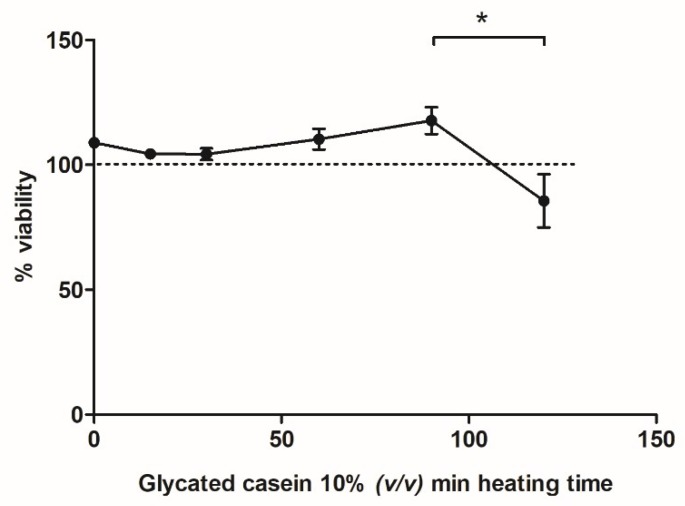
Cell viability of human macrophage-like cells after 6 h exposure to glycated casein from different heating time points (10% *v/v*), measured by the MTT assay. Results were normalized to control (50% *v/v* DPBS), which was set to 100% (dotted line). Data are presented as mean ± SD, *n* = 4, * = *p* < 0.05.

**Figure 5 nutrients-10-01868-f005:**
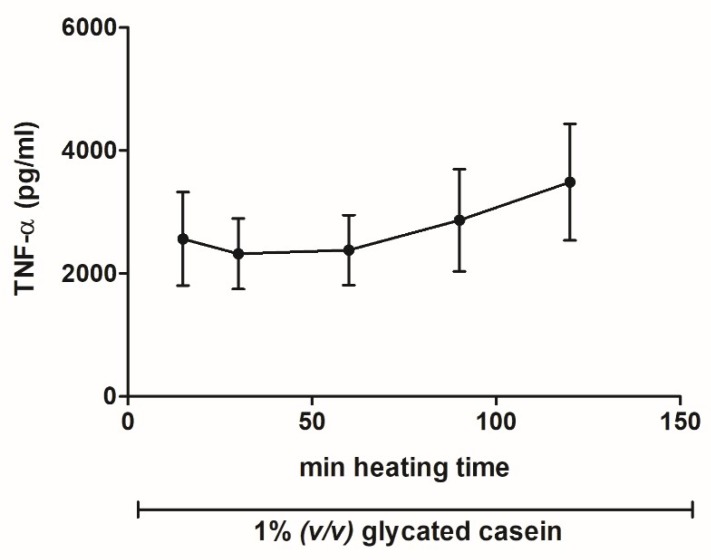
TNF- α secretion of human macrophage-like cells after 6 h exposure to 1% (*v/v*) glycated casein from different heating time points measured by ELISA. Data are presented as mean ± SD, *n* = 4.

**Figure 6 nutrients-10-01868-f006:**
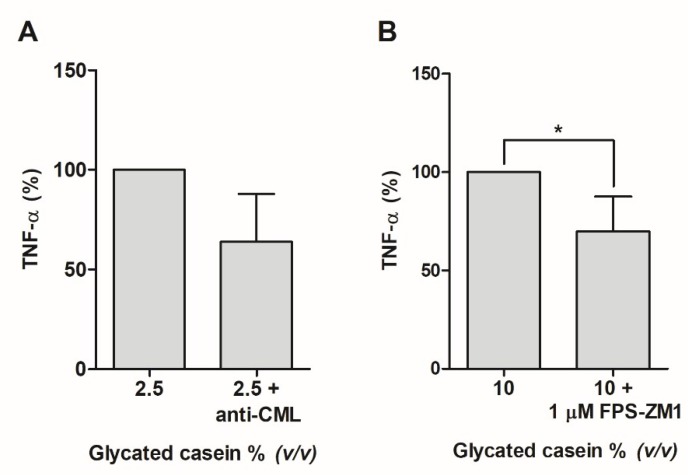
(**A**) TNF-α secretion of human macrophage-like cells after 6-hour exposure to 2.5% (*v/v*) glycated casein with or without anti-CML antibody measured by ELISA. After addition of an anti-CML antibody, TNF-α secretion decreased by 36% ± 24%. Data are presented as mean ± SD, *n* = 3. (**B**) TNF- α secretion of human macrophage-like cells after 6-hour exposure to 10% (*v/v*) glycated casein with or without 1 µM RAGE antagonist (FPS-ZM1) measured by ELISA. After addition of the RAGE blocker, TNF-α secretion decreased by 30 ± 18%. Data are presented as mean ± SD, *n* = 4, * = *p* < 0.05.

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
