# Peer review of "Dietary Advanced Glycation Endproducts Induce an Inflammatory Response in Human Macrophages in Vitro"

_nutrients, 2018, doi:10.3390/nu10121868_

Reviewer 1 Report

The article entitled "Dietary advanced glycation end products induce an inflammatory response in human macrophages invitro" by et al for the first time discovered that dietary AGEs directly stimulate the inflammatory response of human innate immune cells that will define the risk of a regular consumption of AGE-rich food products on human health.

Here are my minor comments for this paper

o   The research design and the experiments are perfect.

o   The paper is well written with very nice discussion.

o   The authors are suggested to introduce more about the inflammatory pathways and the role of TLR adaptor (MyD88) on TNF production. For that the authors are suggested to refer the following paper. https://www.ncbi.nlm.nih.gov/pubmed/30038386  

o   The authors are advised to discuss about the possible pathways of TNF production after RAGE stimulation.

o   The authors are advised to look at the activation status of the NF-kB and MAPK pathways after AGE stimulation in RAW264.7 cells.

Good Luck,

Author Response

We kindly thank the reviewer for the comments and suggestions. We have used these comments to improve our paper. Please find our responses below.

Point 1: The research design and the experiments are perfect.
Point 2: The paper is well written with very nice discussion.

Response 1 and 2:
We appreciate these positive remarks on our manuscript and thank the reviewer for this.

Point 3: The authors are suggested to introduce more about the inflammatory pathways and the role of TLR adaptor (MyD88) on TNF production. For that the authors are suggested to refer the following paper. https://www.ncbi.nlm.nih.gov/pubmed/30038386 

Response 3: We thank the reviewer for this suggestion and the link to this excellent paper. TLR are very important in inflammation caused by LPS, and MyD88 is an important regulator in the activation of NF-κB by TLR. However, AGEs have not been found to bind to TLRs and neither has it been shown that MyD88 interacts with RAGE. We therefore do not think that this information belongs in our paper. We have read the suggested paper carefully and will certainly keep it in mind.

 Point 4: The authors are advised to discuss about the possible pathways of TNF production after RAGE stimulation.

Response 4: In addition to the sentence we already included in the introduction “RAGE activates many enzymes and protein complexes, among which nuclear factor kappa-light-chain-enhancer of activated B cells (NF-κB)” (lines 47-49). We now have included the mechanism of TNF-alpha production through RAGE activation and MAP kinase involvement in our discussion in lines 271-275.

“The most well-known pathway for TNF-α production through RAGE stimulation is NF-κB activation via activation of mitogen-activated protein (MAP) kinases, leading to TNF-α secretion [17]. Indeed, this has been shown by Lander et al. in rat pulmonary smooth muscle cells. It would be of interest to investigate this in human macrophages in future experiments.”

Point 5: The authors are advised to look at the activation status of the NF-kB and MAPK pathways after AGE stimulation in RAW264.7 cells.

Response 5: We thank the reviewer for this valuable comment and have included this mechanism in our discussion (included also in lines 271 and 275), which has already been investigated in smooth muscle cells: http://www.jbc.org/content/272/28/17810.long Unfortunately, the 5 days given for revision gives us too little time to conduct this experiment but will definitely consider this in the future..

We hope that after reading our adjustment, you will find our manuscript acceptable for publication in Nutrients.

Reviewer 2 Report

This is an interesting and well executed study on the effects of dietary AGEs on human macrophages in vitro. This is an important area of clinical research. Main suggestion: The authors should consider whether their hypothesis would be strengthen by referring to a previous study in which dietary AGEs were also found to induce inflammatory markers in vitro but using a different cellular type: HUVEC cells (Cai W et al. Oxidative stress-inducing carbonyl compounds from common foods: novel mediators of cellular dysfunction. Mol Med 2002; 8(7):337-46)

Author Response

Point 1: This is an interesting and well executed study on the effects of dietary AGEs on human macrophages in vitro. This is an important area of clinical research. Main suggestion: The authors should consider whether their hypothesis would be strengthen by referring to a previous study in which dietary AGEs were also found to induce inflammatory markers in vitro but using a different cellular type: HUVEC cells (Cai W et al. Oxidative stress-inducing carbonyl compounds from common foods: novel mediators of cellular dysfunction. Mol Med 2002; 8(7):337-46)

Response 1:

We kindly thank the reviewer for the comments and suggestion to improve our paper. We have now indeed included this study in our discussion, line 260 – 262.

“Interestingly, Cai et al. fractioned AGEs from food products and showed their involvement in intracellular oxidative stress and some inflammatory potential in human umbilical vein endothelial cells. This underlines our results that dietary AGEs induce cellular stress [16].”

We hope that after reading our adjustment, you will find our manuscript acceptable for publication in Nutrients.